

# A new species of freshwater crab of the genus *Qianguimon* Huang, 2018 (Decapoda: Brachyura: Potamidae) from Guangxi, Southern China

Song-Bo Wang[1], Ya-Nan Zhang[1] and Jie-Xin Zou[1,2]

[1] Research Laboratory of Freshwater Crustacean Decapoda & Paragonimus, School of Basic Medical Sciences, Nanchang University, Nanchang City, Jiangxi Province, China
[2] Key Laboratory of Poyang Lake Environment and Resource Utilization, Ministry of Education, Nanchang University, Nanchang City, Jiangxi Province, China

## ABSTRACT

A new species of freshwater crab of the genus *Qianguimon Huang, 2018*, is described from Guangxi Zhuang Autonomous Region, southern China. It can be distinguished from congeners by the following characters: male first gonopods bent inward at about 45° at base of terminal segment, carapace regions distinct and rugged and the female vulva opening inwards and downwards. In addition, molecular evidence derived from the 16S rRNA gene supported the species described in this study as a new species of *Qianguimon*.

## INTRODUCTION

China is the global center of freshwater crab diversity, it has the richest number of freshwater crab species in the world, with more than 300 species from 48 genera and two subfamilies with many more to be discovered (*Dai, 1999*; *Yeo et al., 2008*; *Cumberlidge et al., 2011*; *Chu et al., 2018*; *Chu, Wang & Sun, 2018*; *Huang, Shih & Ahyong, 2018*; *Huang, Wong & Ahyong, 2018*; *Naruse, Chia & Zhou, 2018*; *Wang, Huang & Zou, 2019*; *Wang, Zhou & Zou, 2019*). Also, more than 90% of China's freshwater crab species are distributed in the "China" freshwater zoogeographical subregion (*Chu et al., 2018*; *Huang, Ebach & Ahyong, 2020*).

*Qianguimon* is a genus established by *Huang (2018)*, with four species have been reported at present. The type species *Q. aflagellum* was originally described as *Isolapotamon aflagellum* by *Dai et al. (1980)* from Zhaoping, Guangxi Zhuang Autonomous Region. Afterwards, *Huang (2018)* recorded two additional localities for this species from Mengshan and Chengzhong, Guangxi, and placed it in the genus *Qianguimon*. *Huang (2018)* also reported another two new species of this genus: *Q. splendidum* from Yanghe, Guangxi and *Q. elongatum* from Leishan, Guizhou Province. *Wang, Huang & Zou (2019)* subsequently described the fourth species: *Q. rongxianense* from Rong, Guangxi. The prominent feature of this genus is the boot-shaped terminal

Corresponding author
Jie-Xin Zou, jxzou@ncu.edu.cn

segment of the male first gonopod (*Huang, 2018*; *Wang, Huang & Zou, 2019*). They have a broad altitude range, from close to sea level to over 1,000 m, and can be found at altitudes as high as 1,500 m (*Huang, 2018*).

Species exploration is ongoing. In a joint research survey with Chao Huang and Si-Ying Mao, we discovered a new species of the genus *Qianguimon* from Yuzhou District, Yulin City, Guangxi Zhuang Autonomous Region, southern China during. It is herein described as a new species.

## MATERIALS AND METHODS

### Material examined

Specimens were collected from Yuzhou District of Yulin City in Guangxi Zhuang Autonomous Region by Song-Bo Wang, preserved in 95% ethanol; and deposited at the Department of Parasitology of the Medical College of Nanchang University, Jiangxi, China (NCU MCP), National Tropical Disease Research Center, Shanghai, China (TDRC), Zoological Reference Collection of the Raffles Museum of Biodiversity Research, National University of Singapore, Singapore (ZRC), Sun Yat-sen Museum of Biology, Sun Yat-sen University, Guangzhou, China (SYSBM). Some of the comparative materials were also deposited at the Sun Yat-sen Museum of Biology, Sun Yat-sen University, Guangzhou, China (SYSBM). Carapace width and length were measured in millimeters. The abbreviations G1 and G2 refer to the male first and second gonopods, respectively. The terminology used herein primarily follows that of *Dai (1999)* and *Davie, Guinot & Ng (2015)*.

### Molecular analyses

Muscle tissue was excised from chelipeds, total genomic DNA was extracted from the tissue using the Omega Tissue Kit following the manufacturer's protocol. Then, the 16S rRNA gene was amplified using polymerase chain reaction (PCR) with the primers 1471 (5′-CCTGTTTANCAAAAACAT-3′) and 1472 (5′-AGATAGAAACCAACCTGG-3′) (*Crandall & Fitzpatrick, 1996*). The PCR conditions were as follows: denaturation for 50 s at 94 °C, annealing for 40 s at 52 °C and extension for 1 min at 72 °C (33 cycles), followed by a final extension for 10 min at 72 °C. The PCR products were purified and sequenced using an AB I3730 automatic sequencer.

We performed the molecular analysis with the mitochondrial 16S rRNA gene fragment. In total, 26 species of 18 genera were used to construct phylogenetic trees (Table 1). Sequences were aligned using MAFFT ver.7.215 (*Katoh & Standley, 2013*) based on the G-INS-I method, and the conserved regions were selected with Gblocks 0.91b (*Castresana, 2000*) using the default settings. The best-fitting model for Bayesian Inference (BI) analysis was determined by MrModeltest ver.2.2 (*Nylander, 2005*), selected by the Akaike information criterion (AIC). The obtained model was GTR+I+G (*Tavaré, 1986*). MrBayes ver.3.2.6 (*Ronquist et al., 2012*) was employed to perform the BI analysis, and four Monte Carlo Markov Chains of 2,000,000 generations were run with sampling every 1,000 generations. The first 500,000 generations were discarded as burn-in. The best evolutionary model for Maximum Likelihood (ML) analysis was HKY+I+G

**Table 1 GenBank accession numbers of the species used for phylogenetic analysis.** The 16S rRNA genes of 26 species belonging to 18 genera of the subfamily Potamidae from Asia.

| Species | Museum number | Locality | GenBank number | References |
|---|---|---|---|---|
| *Aparapotamon grahami* Rathbun, 1929 | ZRC 0334(II) | Yunnan, China | AB428489 | *Shih, Yeo & Ng (2009)* |
| *Apotamonautes hainanensis* Parisi, 1916 | ZRC | Hainan, China | AB428459 | *Shih, Yeo & Ng (2009)* |
| *Chinapotamon glabrum* Dai et al., 1980 | CAS CB | Guangxi, China | AB428451 | *Shih, Yeo & Ng (2009)* |
| *Cryptopotamon anacoluthon* Kemp, 1918 | NCHUZOOL 13122 | Hong Kong | AB428453 | *Shih, Yeo & Ng (2009)* |
| *Daipotamon minos* Ng & Trontelj, 1996 | ZRC | Guizhou, China | LC198524 | *Huang, Shih & Ng (2017)* |
| *Diyutamon cereum* Huang, Shih & Ng, 2017 | SYSBM | Guizhou, China | LC198520 | *Huang, Shih & Ng (2017)* |
| *Hainanpotamon fuchengense* Dai, 1995 | NCHUZOOL 13128 | Hainan, China | AB428461 | *Shih, Yeo & Ng (2009)* |
| *Longpotamon baiyanense* Ng & Dai, 1997 | ZRC | Hunan, China | AB428470 | *Shih, Yeo & Ng (2009)* |
| *Mediapotamon leishanense* Dai, 1995 | SYSBM 001094 | Guizhou, China | LC155164 | *Shih, Huang & Ng (2016)* |
| *Mediapotamon liboense* Wang & Zhou, 2019 | NCU MCP 343004 | Guizhou, China | MK820377 | *Wang, Zhou & Zou (2019)* |
| *Neotiwaripotamon jianfengense* Dai & Naiyanetr, 1994 | NCHUZOOL 13127 | Hainan, China | AB428460 | *Shih, Yeo & Ng (2009)* |
| *Parapotamon spinescens* Calman, 1905 | NCU MCP | Yunnan, China | AB428467 | *Shih, Yeo & Ng (2009)* |
| *Pararanguna semilunatum* Dai & Chen, 1985 | ZRC | Yunnan, China | AB428490 | *Shih, Yeo & Ng (2009)* |
| *Potamiscus yongshengense* Dai & Chen, 1985 | NNU150951 | Yunnan, China | KY963597 | *Chu, Zhou & Sun (2017)* |
| *Qianguimon splendidum* Huang, 2018 | SYSBM 001598 | Guangxi, China | MG709241 | *Huang (2018)* |
| *Qianguimon aflagellum* Dai et al., 1980 | SYSBM 001404 | Guangxi, China | MG709239 | *Huang (2018)* |
| *Qianguimon elongatum* Huang, 2018 | SYSBM 001424 | Guizhou, China | MG709240 | *Huang (2018)* |
| *Qianguimon rongxianense* Wang, 2019 | NCU MCP 118401 | Guangxi, China | MK335483 | *Wang, Huang & Zou (2019)* |
| *Socotrapotamon nojidensis* Apel & Brandis, 2000 | ZRC 2000.2232 | Socotra, Yemen | AB428493 | *Shih, Yeo & Ng (2009)* |
| *Tenuipotamon huaningense* Dai & Bo, 1994 | CAS CB 05175 | Yunnan, China | AB428491 | *Shih, Yeo & Ng (2009)* |
| *Trichopotamon daliense* Dai & Chen, 1985 | NCHUZOOL 13130 | Yunnan, China | AB428492 | *Shih, Yeo & Ng (2009)* |
| *Yarepotamon gracilipa* Dai et al., 1980 | ZRC | Guangxi, China | AB428452 | *Shih, Yeo & Ng (2009)* |
| *Yarepotamon fossor* Huang, 2018 | SYSBM 001417 | Guangxi, China | MG709238 | *Huang (2018)* |
| *Yarepotamon breviflagllum* Dai & Tüerkay, 1997 | SYSBM 001442 | Guangdong, China | MG709236 | *Huang (2018)* |
| *Yarepotamon meridianum* Huang, 2018 | SYSBM 001581 | Guangdong, China | MG709237 | *Huang (2018)* |
| *Qianguimon yuzhouense* n. sp. | NCU MCP 415701 | Guangxi, China | MN844075 | This study |
| *Qianguimon yuzhouense* n. sp. | NCU MCP 415704 | Guangxi, China | MN844076 | This study |
| *Qianguimon yuzhouense* n. sp. | NCU MCP 415705 | Guangxi, China | MN844077 | This study |

Note:
CAS CB, Chinese Academy of Sciences, Beijing, China; NCHUZOOL, Zoological Collections of the Department of Life Science, National Chung Hsing University, Taichung, Taiwan; NCU MCP, Department of Parasitology of the Medical College of Nanchang University, Jiangxi, China; NNU, College of Life Sciences, Nanjing Normal University, Nanjing, China; SYSBM, Sun Yat-sen Museum of Biology, Sun Yat-Sen University, Guangzhou, China; ZRC, Zoological Reference Collection of the Raffles Museum of Biodiversity Research, National University of Singapore, Singapore.

(*Hasegawa, Kishino & Yano, 1985*), determined by MEGA ver.X.0 (*Kumar et al., 2018*) based on the Bayesian information criterion (BIC). The ML tree was built based on 1,000 bootstrap replicates in MEGA ver.X.0 (*Kumar et al., 2018*). The pairwise estimates of Kimura 2-parameter (K2P) distances (*Kimura, 1980*) among the five species of *Qianguimon* were calculated using MEGA ver.X.0 (*Kumar et al., 2018*).

The electronic version of this article in portable document format will represent a published work according to the International Commission on Zoological Nomenclature (ICZN), and hence the new names contained in the electronic version are effectively published under that Code from the electronic edition alone. This published work and the nomenclatural acts it contains have been registered in ZooBank, the online registration system for the ICZN. The ZooBank LSIDs (Life Science Identifiers) can be resolved and the associated information viewed through any standard web browser by appending the LSID to the prefix http://zoobank.org/. The LSID for this publication is: urn:lsid: zoobank.org:pub:7BFE0C18-76EE-483C-9B5F-C0143C5B6A16. The online version of this work is archived and available from the following digital repositories: Peer J, PubMed Central, and CLOCKSS.

## RESULTS

### Systematics

**Family Potamidae Ortmann, 1896**
*Qianguimon Huang, 2018*

***Qianguimon yuzhouense* n. sp.** (Figs. 1–4)
urn:lsid:zoobank.org:act: A785F440-CFB0-42A8-9304-7433E6FE57A8

**Material examined.** Holotype: male (21.3 × 18.2 mm) (NCU MCP 415701), Winding road beside Hanshan Temple on Gui Mountain (22°41′5.18″N 110°12′58.56″E, 246 m asl.), Yuzhou District, Yulin City, Guangxi Zhuang Autonomous Region, China, coll. Song-Bo Wang, Jie-Xin Zou, Chao Huang, Si-Ying Mao, 18 Dec. 2018. Paratypes: 2 males (18.7 × 16.0 mm, 20.3 mm × 16.6 mm) (TDRC 002003, ZRC 2019.1662), 2 females (14.5 × 12.3 mm, 14.6 × 12.1 mm) (NCU MCP 415703, TDRC 002004), same data as holotype. Others: 5 males (16.0 × 13.8 mm, 15.7 × 13.4 mm, 22.6 × 18.9 mm, 19.5 × 16.9 mm, 14.1 × 11.5 mm; NCU MCP 415704, NCU MCP 415705, SYSBM 001977, SYSBM 001978, SYSBM 001979) and 1 female (15.8 × 13.2 mm) (SYSBM 001980), same data as holotype.

**Comparative material.** *Qianguimon rongxianense Wang, Huang & Zou, 2019*: Holotype: 1 male (15.2 × 12.8 mm) (NCU MCP 118401), Sixian Village, Licun Town, Rong County, Yulin City, Guangxi Zhuang Autonomous Region, small stream, coll. Ye-Song Cheng, August 23, 2007; Paratype, 1 female (allotype) (20.4 × 16.0 mm) (NCU MCP 118403), same data as holotype. *Qianguimon aflagellum Huang, 2018*: 1 male (19.4 × 15.8 mm) (SYSBM 001403), Wuzhou, Mengshan, Guangxi Province, shallow creek, April 2014 coll. C. Huang; 1 female (22.7 × 18.0 mm) (SYSBM 001404), same data as above [photos examined]. *Qianguimon elongatum Huang, 2018*: Holotype, 1 male (22.0 × 16.8 mm) (SYSBM 001421), Leishan County, Qiandongnan Miao and Dong Autonomous Prefecture, Guizhou Province, mud burrows at the side of hillstreams, July 2013, coll. C. Huang; Paratypes, 1 female (allotype), (29.0 × 21.5 mm) SYSBM 001423, same data as holotype (photos examined). *Qianguimon splendidum Huang, 2018*: Holotype, 1 male

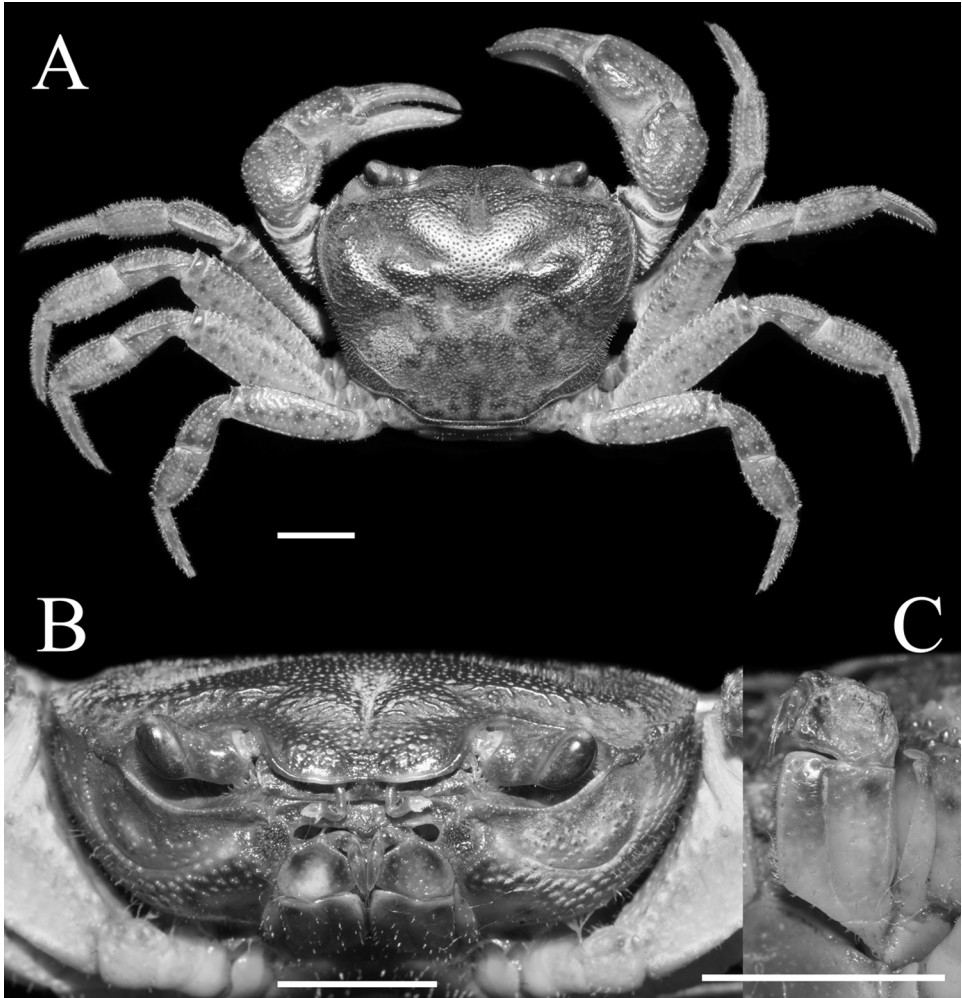

**Figure 1** *Qianguimon yuzhouense* n. sp. Holotype male (21.3 × 18.2 mm) (NCU MCP 415701). (A) Overall habitus; (B) frontal view of the cephalothorax; (C) left third maxilliped. Scales = 5 mm. Photo credit: Song-Bo Wang.

(27.8 × 21.1 mm) (SYSBM 001597), Yanghe County, Liuzhou City, Guangxi Zhuang Autonomous Region, mud burrows at the side of hillstreams, September 2015, coll. C. Huang; Paratype, 1 female (allotype) (30.8 × 23.0 mm) (SYSBM 001598), same data as holotype (photos examined).

**Diagnosis.** Carapace broader than long, regions distinct, anterolateral regions rugose; cervical groove and H-shaped groove deep, distinct; epigastric cristae conspicuous, postorbital cristae sharp. External orbital angle narrowly triangular, separated from anterolateral margin by gap; epibranchial teeth distinct; anterolateral margin lined with conspicuous granules. Third maxilliped merus median depression indistinct, exopod with vestigial flagellum. Chelipeds slightly unequal; outer surfaces of chelae smooth; fingers with very small gap when closed. Male pleon narrowly triangular, lateral margins gently concave; telson triangular, somite 6 transversely trapeziform. Male sterno-pleonal cavity very deep, median longitudinal suture of sternites 7/8 deep and relatively long.

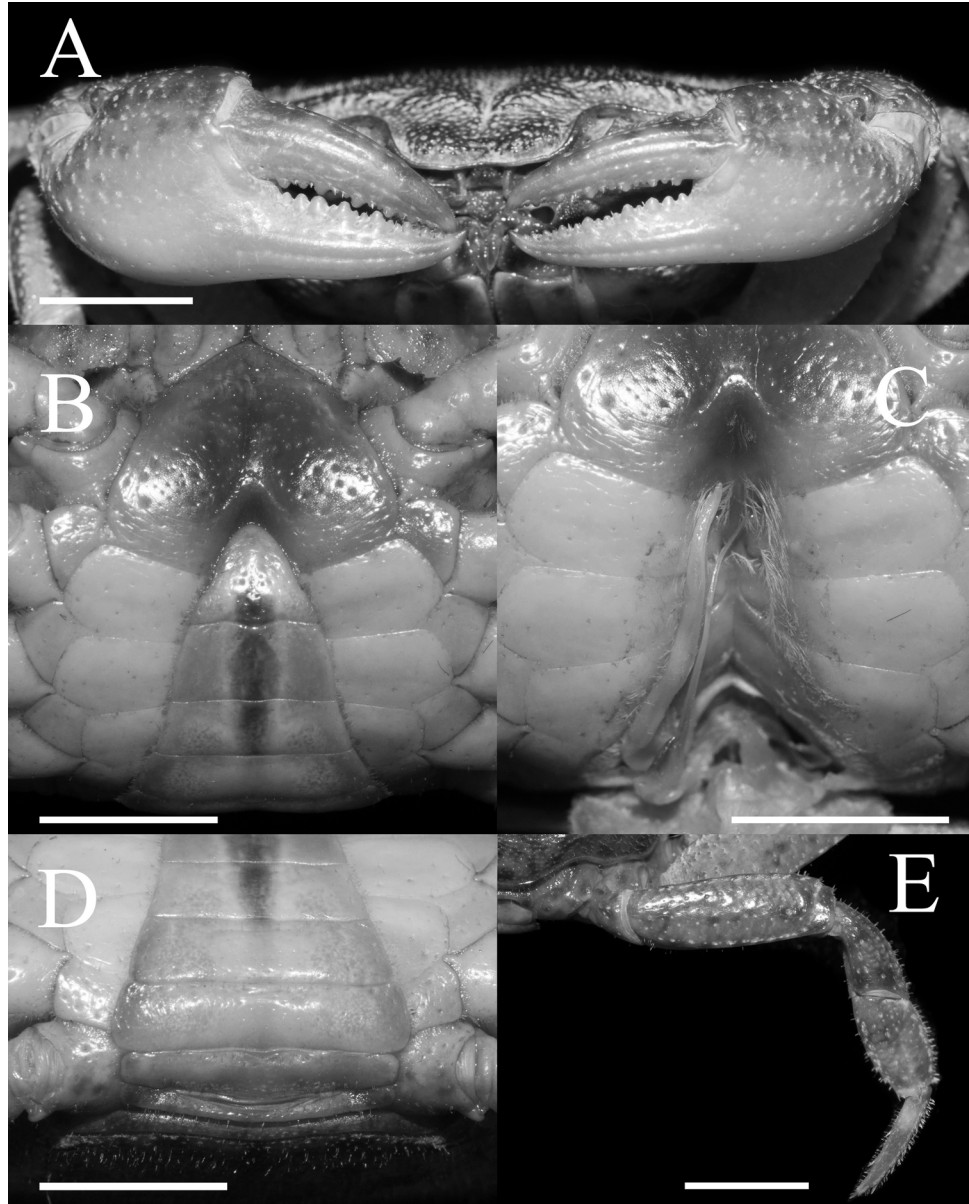

**Figure 2** *Qianguimon yuzhouense* **n. sp. Holotype male (21.3 × 18.2 mm) (NCU MCP 415701).**
(A) Outer view of chelipeds; (B) ventral view of anterior thoracic sternum, telson, and male pleonal somites 4–6; (C) ventral view of sterno-pleonal cavity with G1 in situ; (D) male pleonal somites 1–4; (E) the fourth ambulatory leg. Scales = 5 mm. Photo credit: Song-Bo Wang.

G1 very slender, terminal segment boot-shaped, distinctly sinuous, tip of terminal segment exceeding sternites 4/5 suture. Female vulva reaching sternites 5/6 suture, with opening directed inward at an angle of 45°.

**Description.** Carapace sub-quadrate, 1.1-1.2 times as broad as long (mean = 1.18); regions distinct, dorsal surface with pits and scattered setae; anterolateral region wrinkled (Figs. 1A and 3A). Branchial regions slightly swollen. Cervical groove very deep, distinct;

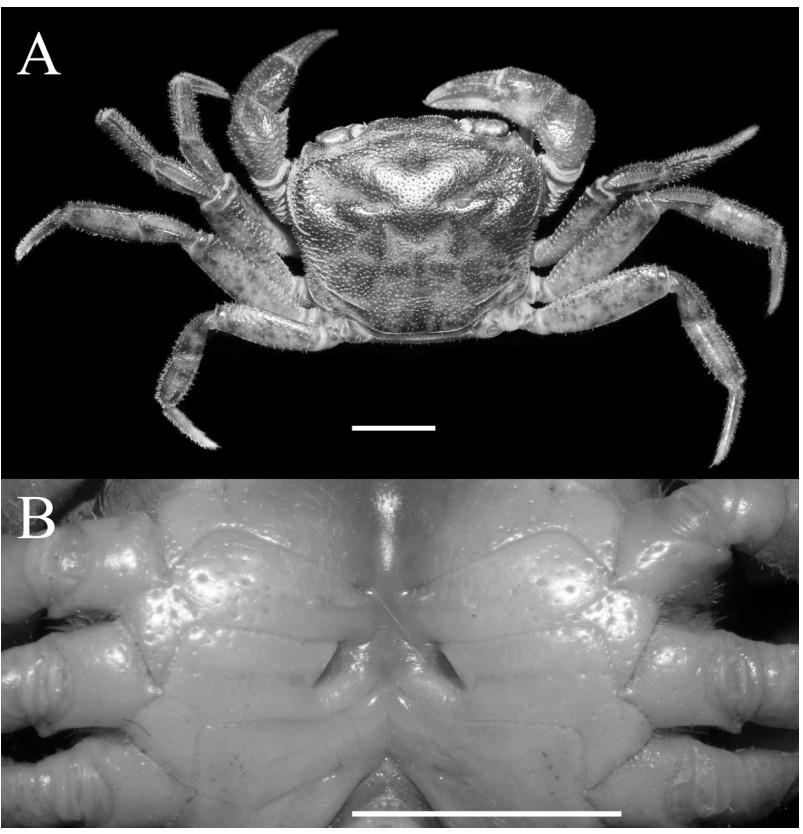

**Figure 3** *Qianguimon yuzhouense* n. sp. Paratype female (14.5 × 12.3 mm) (NCU MCP 415703). (A) Overall habitus; (B) female vulvae. Scales = 5 mm. Photo credit: Song-Bo Wang.

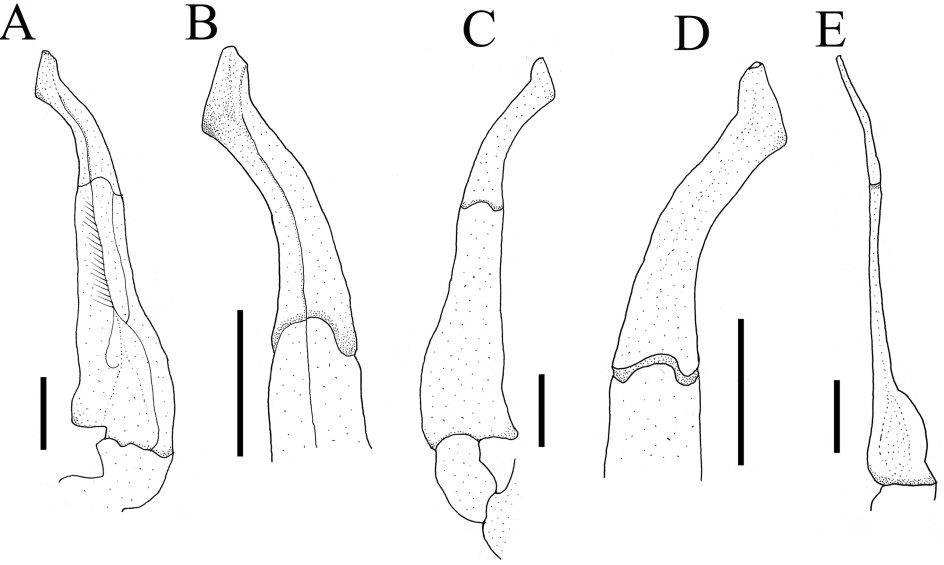

**Figure 4 Gonopods of holotype.** (A) Ventral view of the left G1; (B) ventral view of the terminal segment of left G1; (C) dorsal view of the left G1; (D) dorsal view of the terminal segment of left G1; (E) ventral view of the left G2. Scales = 1 mm. Photo credit: Song-Bo Wang.

H-shaped groove between gastric and cardiac regions deep and distinct (Figs. 1A and 3A). Epigastric cristae conspicuous, separated by narrow gap; postorbital cristae sharp, not fuzed with epigastric cristae, nearly reaching the anterolateral margin (Figs. 1A and 3A). Front distinctly deflexed, margin ridged in dorsal view, medially concave (Figs. 1A and 3A). External orbital angle narrowly triangular, very sharp, margins smooth and without any granules, separated from anterolateral margin by small distinct V-shaped gap; epibranchial teeth small, distinct, granular (Figs. 1A and 3A). Anterolateral margin distantly cristate, lined with approximately 16 granules, lateral part bent inward; posterolateral surface smooth, with inconspicuous oblique striae, converging towards posterior carapace margin (Figs. 1A and 3A). Orbits medium size; supraorbital, infraorbital margins cristate, smooth and without granules (Fig. 1B). Sub-orbital regions covered with scattered rounded granules; sub-hepatic regions and pterygostomial regions covered numerous large granules (Fig. 1B). Epistome posterior margin narrow; median lobe triangular, lateral margins oblique (Fig. 1B).

Third maxilliped merus about 1.3 times as broad as long, trapezoidal, median depression indistinct; ischium about 1.5 times as long as broad, rectangular, with distinct median sulcus; exopod reaching approximately 1/5 of merus length, with vestigial flagellum; dactylus not reaching the upper edge of ischium (Fig. 1C).

Chelipeds slightly unequal (Fig. 2A). Merus cross-section trigonal, with inner-lower margin crenulated (Fig. 2A). Carpus surface weakly wrinkled, with prominent sharp spine at inner-distal margin (Fig. 1A). Outer surfaces of chelae pitted, palm of larger chela about 1.2 times as long as high (Fig. 2A). Movable finger approximately as long as the immovable finger; inner margin of fingers with numerous round and blunt teeth; fingers forming inconspicuous gap when closed (Fig. 2A).

Male thoracic sternum generally smooth, pitted (Fig. 2B). Sternites 1, 2 completely fuzed to form triangular structure; sternites 2, 3 separated by obvious suture; sternites 3, 4 fuzed (Fig. 2A). Male sterno-pleonal cavity very deep, nearly reaching imaginary line connecting mid-length of cheliped coxae (Fig. 2C). Median longitudinal suture of sternites 7, 8 deep and relatively long; male pleonal locking tubercle inconspicuous, round, on posterior third of sternite 5 (Fig. 2C). Female vulva reaching sternites 5/6, reaching proximal three-quarters width of sternite 6; upper and lower margin flat without any swelling; opening directed inward at about an angle of 45° (Fig. 3B).

Male pleon narrowly triangular, lateral margins gently concave; telson triangular, lateral margins oblique, straight; somite 6 transversely trapeziform, about 2.1 times as broad as long (Fig. 2B); somites 3–5 trapezoidal, gradually decreasing in width, increasing in length, lateral margins oblique; somites 1 and 2 sub-rectangular, very wide, the former flatter, reaching to bases of coxae of fourth ambulatory legs (Fig. 2D).

Ambulatory legs slender; the second pair longest and last pair shortest (Fig. 1A). Merus longest, without subdistal spine or tooth; carpus stout, dorsal margin with cristae (Fig. 1A). The fourth leg propodus about 1.9 times as long as broad, slightly shorter than dactylus; dactylus sharp, with several spines and numerous setae on the surface (Fig. 2E).

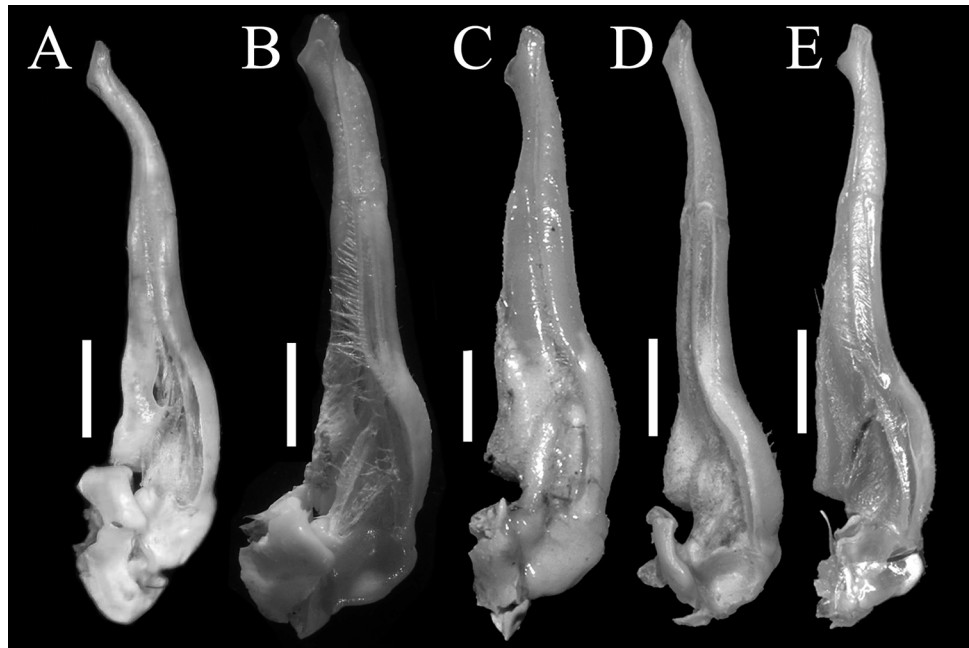

**Figure 5 The left G1s of the five species of *Qianguimon*.** (A) *Q. yuzhouense* n. sp., NCU MCP 415701; (B) *Q. rongxianense Wang, Huang & Zou, 2019*, NCU MCP 118401; (C) *Q. aflagellum*, *Huang, 2018*, SYSBM 0014033; (D) *Q. elongatum*, *Huang, 2018*, SYSBM 001421 dorsal view of the terminal segment of right G1; (E) *Q. splendidum*, *Huang, 2018*, SYSBM 001597. Scales = 1 mm. Photo credit: Chao Huang.

G1 very slender, dorsal and ventral surface smooth, lateral margin without seta, terminal segment boot-shaped, distinctly sinuous, bend inward at a 45° angle medially, with blunt sub-distal projection (Figs. 4A–4D and 5A); tip of terminal segment exceeding sternites 4/5 suture (Fig. 2D); subterminal segment about 2.0 times as long as terminal segment. G2 elongate, almost equal to G1 in length; basal segment about 2.2 times length of distal segment, basal segment sub-ovate (Fig. 4E). Groove for G2 located medially on the ventral side of G1 subterminal segment, thin setae on distal regions of G1 subterminal segment (Fig. 4A).

**Remarks.** The new species is similar to other species *Qianguimon*, in its carapace broader than long, postorbital and epigastric cristae visible; exopod of the third maxilliped with short or no flagellum, male pleon triangular; G1 generally slender, terminal segment boot-shaped with sub-distal projection; vulvae medium-sized and reaching proximal three-quarters width of sternite 6. But *Q. yuzhouense* n. sp. can be differentiated from congeners by its regions distinct and dorsal surface rugged, narrowly triangular and sharp external orbital angle, blunt and broadly triangular epibranchial tooth, G1 very slender and bent inward at about 45° at base of terminal segment, tip exceeding sternites 4/5 suture in situ, female vulva opening inward at a 45° angle. Other differences are listed in Table 2 and Fig. 5.

**Etymology.** The new species is named after the type locality, Yuzhou District, Yulin City, Guangxi Zhuang Autonomous Region, China.

**Table 2 Morphological differences between the five species of *Qianguimon Huang, 2018*.**

| Species/character | *Q. yuzhouense* n. sp. | *Q. rongxianense* | *Q. aflagellum* | *Q. elongatum* | *Q. splendidum* |
|---|---|---|---|---|---|
| Carapace | Regions distinct, surface rugged | Regions indistinct, surface generally smooth | Regions indistinct, surface generally smooth | Regions indistinct, surface generally smooth | Regions indistinct, surface very smooth |
| Flagellum of exopod of third maxilliped | Very short to absent | Short length | Very short to absent | Absent | Absent |
| G1 in situ | Exceeding sternites 4/5 suture | Not reaching sternites 4/5 suture | Reaching to sternites 4/5 suture | Well exceeding sternites 4/5 suture | Exceeding sternites 4/5 suture |
| G1 and the shape of sub-distal projection | Very slender, bend inward about 45°, blunt | Generally slender, bend inward about 20°, large triangular | Generally slender, upward straightly, large triangular | Very slender, bend inward about 20°, blunt | Very slender, upward straightly, large triangular |
| Opening of female vulvae | Inward and deflect about 45° | Inward and deflect about 20° | Inward without deflect | Inward without deflect | Inward and deflect about 20° |

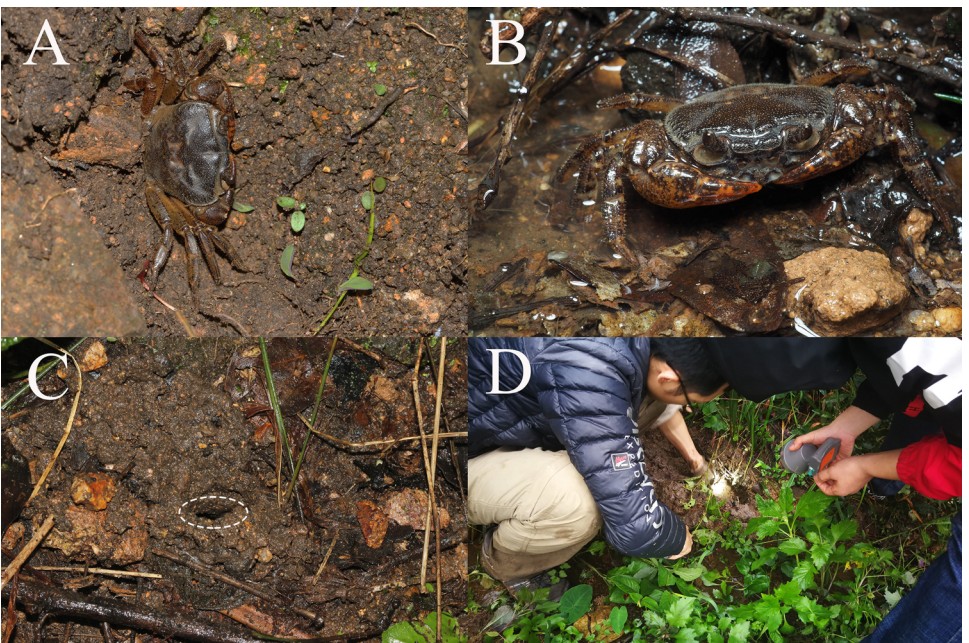

**Figure 6 Habitat environment.** (A and B) Color in life; (C) a burrow inhabited by the new species (indicated by circle); (D) collecting specimens by hand. Photo credit: Chao Huang.

**Living color.** Most of the carapace is dark brown. The chelipeds are brown to orange, while the ambulatory legs are brown. The overall color is consistent with the surrounding environment (Figs. 6A and 6B).

**Ecology.** This species was found in a stream next to a mountain road. The stream has no obvious flowing water, and has lush weeds growing in it. We found the crab burrows by removing the weeds. The burrows are sandy and without much soil. We found the crabs after digging about 10 cm into the burrows (Figs. 6C and 6D).

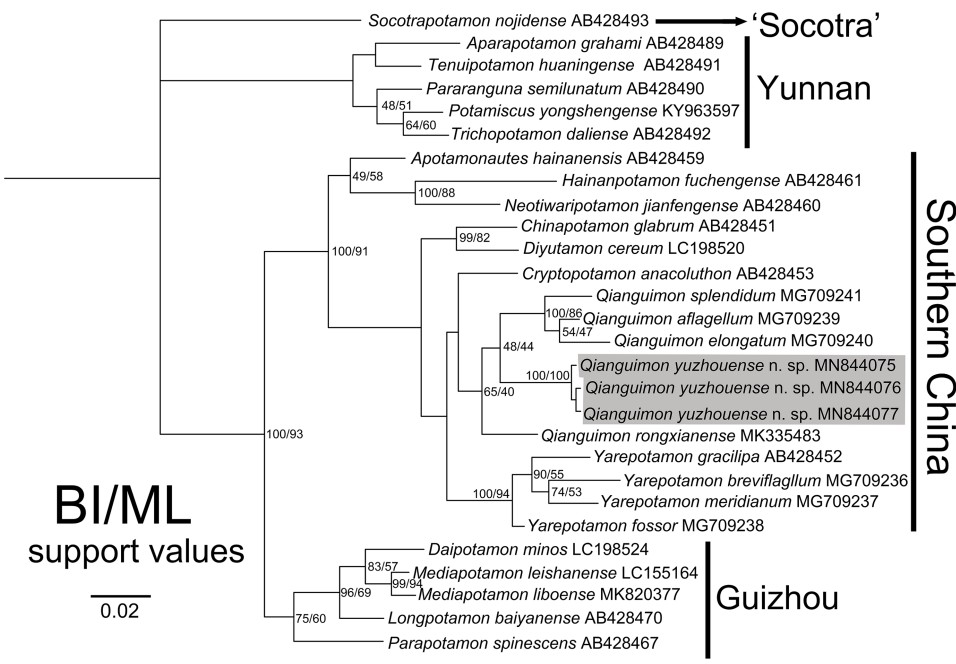

**Figure 7 Phylogenetic tree.** Based on the 16S rRNA genes of the *Qianguimon yuzhouense* n. sp. and some other species for comparison. Topologies and branch lengths were obtained from BI analysis. Support values represented at the nodes were from BI and ML. Photo credit: Song-Bo Wang.

## Phylogenetic analyses

In this study, we obtained the 16S rRNA molecular data of three specimens collected from Yuzhou District, Yulin City, Guangxi Zhuang Autonomous Region, China. The alignment sequences were downloaded from GenBank and include 26 species from 18 genera of the subfamily Potamidae Ortmann, 1896 from Asia. The access numbers can be found in Table 1. We used the BI and ML methods to construct the phylogenetic tree. The topological structure of the trees showed a high degree of consistency (Fig. 7). The three mitochondrial 16S rRNA gene fragments of the new species are very close genetically, with the pairwise genetic distances zero (Table 3), which indicates that they are sequences from the same species and are consistent with the results of the morphological study. The new species are clustered together with *Q. rongxianense*, *Q. aflagellum*, *Q. elongatum* and *Q. splendidum*, and form an independent branch in the clade "Southern China" (*Huang, Ebach & Ahyong, 2020*), indicating that the five species are congeners of the genus *Qianguimon*. The minimum interspecific pairwise K2P genetic distances of the new species and other congeners is 0.037507 (Table 3).

## DISCUSSION

*Qianguimon yuzhouense* n. sp. has the diagnostic features of *Qianguimon*, such as visible postorbital and epigastric cristae and male first gonopod generally slender with boot-shaped terminal segment (*Huang, 2018*). In this study, we collected mitochondrial 16S rRNA gene molecular data for all species of the genus, and based on this, established BI

**Table 3 K2P divergences between sequences of the five species from *Qianguimon Huang, 2018*.**

|  | 1 | 2 | 3 | 4 | 5 | 6 | 7 |
|---|---|---|---|---|---|---|---|
| 1. *Q. splendidum* MG709241 |  |  |  |  |  |  |  |
| 2. *Q. aflagellum* MG709239 | 0.020900 |  |  |  |  |  |  |
| 3. *Q. elongatum* MG709240 | 0.032785 | 0.020900 |  |  |  |  |  |
| 4. *Q. rongxianense* MK335483 | 0.039934 | 0.037587 | 0.037507 |  |  |  |  |
| 5. *Q. yuzhouense* n. sp. MN844075 | 0.040026 | 0.037587 | 0.037507 | 0.040084 |  |  |  |
| 6. *Q. yuzhouense* n. sp. MN844076 | 0.040026 | 0.037587 | 0.037507 | 0.040084 | 0.000000 |  |  |
| 7. *Q. yuzhouense* n. sp. MN844077 | 0.040026 | 0.037587 | 0.037507 | 0.040084 | 0.000000 | 0.000000 |  |

and ML phylogenetic trees. Phylogenetic analysis showed that the five species of the genus formed an independent branch. Both phylogenetic tree and genetic distances suggest that *Q. yuzhouense* is a new species. There are three clades within *Qianguimon*, however, support for these clades is not high. Considering the shared generic characters of these species, we believe that these species all belong to the same genus. The new species is found in Yuzhou District of Yulin City, Guangxi Zhuang Autonomous Region, which is within the distribution of *Qianguimon*. The other four reported species of this genus are all distributed in southern Guizhou Province or eastern Guangxi Zhuang Autonomous Region (*Huang, 2018*; *Wang, Huang & Zou, 2019*). In summary, the species reported in this paper is a new species of *Qianguimon* that is supported by molecular data, morphology and biogeography.

## CONCLUSIONS

In this article, we reported a new species of freshwater crab from Yuzhou District, Yulin City, Guangxi Zhuang Autonomous Region, China. We found that it fits well within the definition of *Qianguimon Huang, 2018*, morphologically, and our molecular analysis also supports it as a new species of the genus *Qianguimon*.

## ACKNOWLEDGEMENTS

We would like to thank Chao Huang and Si-Ying Mao for locating the crabs, assisting us in collecting specimens and for taking photos of live specimens for us to use. We also would like to thank Milan Koch and other two anonymous reviewers for greatly improving this manuscript.

### Funding

This work was supported by the National Parasitic Resources Center (NPRC-2019-194-30), the National Natural Science Foundation of China (Nos. 31560179 and 21866020), and the Nanchang University College Students' Innovation and Entrepreneurship Training Program (No. 2018388) and Nanchang University's Scientific Research Training Program (20190402197). The funders had no role in study design, data collection and analysis, decision to publish, or preparation of the manuscript.

## Grant Disclosures

The following grant information was disclosed by the authors:

National Parasitic Resources Center: NPRC-2019-194-30.

National Natural Science Foundation of China: 31560179 and 21866020.

Nanchang University College Students' Innovation and Entrepreneurship Training Program: 2018388.

Nanchang University's Scientific Research Training Program: 20190402197.

## Competing Interests

The authors declare that they have no competing interests.

## Author Contributions

- Song-Bo Wang conceived and designed the experiments, performed the experiments, analyzed the data, prepared figures and/or tables, and approved the final draft.
- Ya-Nan Zhang conceived and designed the experiments, performed the experiments, analyzed the data, prepared figures and/or tables, and approved the final draft.
- Jie-Xin Zou conceived and designed the experiments, performed the experiments, analyzed the data, authored or reviewed drafts of the paper, and approved the final draft.

## Data Availability

The sequences described in this study are available at GenBank: MN844075 to MN844077.

Specimens are available at the Department of Parasitology of the Medical College of Nanchang University, Jiangxi, China (NCU MCP), National Tropical Disease Research Center, Shanghai, China (TDRC), Zoological Reference Collection of the Raffles Museum of Biodiversity Research, National University of Singapore, Singapore (ZRC), Sun Yat-sen Museum of Biology, Sun Yat-sen University, Guangzhou, China (SYSBM). And the accession numbers for each specimen were as follows: Holotype: male (21.3 × 18.2 mm) (NCU MCP 415701); Paratypes: 2 males (18.7 × 16.0 mm, 20.3 mm × 16.6 mm) (TDRC 002003, ZRC 2019.1662), 2 females (14.5 × 12.3 mm, 14.6 × 12.1 mm) (NCU MCP 415703, TDRC 002004); Others: 5 males (16.0 × 13.8 mm, 15.7 × 13.4 mm, 22.6 × 18.9 mm, 19.5 × 16.9 mm, 14.1 × 11.5 mm; NCU MCP 415704, NCU MCP 415705, SYSBM 001977, SYSBM 001978, SYSBM 001979) and 1 female (15.8 × 13.2 mm) (SYSBM 001980).

## New Species Registration

The following information was supplied regarding the registration of a newly described species:

Publication LSID: urn:lsid:zoobank.org:pub:7BFE0C18-76EE-483C-9B5F-C0143C5B6A16.

*Qianguimon yuzhouense* n. sp. LSID: urn:lsid:zoobank.org:act: A785F440-CFB0-42A8-9304-7433E6FE57A8.
## Supplemental Information

Supplemental information for this article can be found online at http://dx.doi.org/10.7717/peerj.9194#supplemental-information.

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
