# Peer review of "A new species of freshwater crab of the genus Qianguimon Huang, 2018 (Decapoda: Brachyura: Potamidae) from Guangxi, Southern China"

_PeerJ, doi:10.7717/peerj.9194_

## Round 0.1 · original submission · Major Revisions

Your manuscript is not acceptable for publication in its present form.
However, if you feel that you can suitably address the reviewers' comments (included), I invite you to revise and resubmit your manuscript.

The discussion section has to be improved, you could use some new references in order to have more arguments to reinforce your results.

Best regards,


Reviewer 1 ·

Basic reporting

no comment

Experimental design

no comment

Validity of the findings

no comment

Additional comments

Authors has reported A new species of freshwater crab of the genus
Qianguimon, as the fifth species of this genus. The results have been well present. Moreover, molecular analysis added to morphological results has been validated finding. However, in the discussion section, i observed no discussion about findings. I have submitted a series of corrections in the attached version of paper. I emphasis that the discussion section should revise according the comments of attached version. Finally I recommend to publish the article after a major revisions.

Annotated reviews are not available for download in order to protect the identity of reviewers who chose to remain anonymous.

Reviewer 2 ·

Basic reporting

The authors used compound sentences in the parts of living color and ecology. These parts should be rewritten and checked by a native speaker of English.

Experimental design

No comment.

Validity of the findings

No comment.

Additional comments

The main comment is the author should show the DNA results, such as the minimum interspecific divergence between the new species and other species of this genus. And provide a table of genetic distances within the new species and between five species of Qianguimon.

Annotated reviews are not available for download in order to protect the identity of reviewers who chose to remain anonymous.

·

Basic reporting

Comments in attached pdf.

Experimental design

Comments in attached pdf.

Validity of the findings

Comments in attached pdf.

Additional comments

Comments in attached pdf.

---

## Round 0.2 · Minor Revisions

Your manuscript needs some minor modifications which are marked in the annotated manuscript provided.

Reviewer 1 ·

Basic reporting

no comment

Experimental design

no comment

Validity of the findings

no comment

Annotated reviews are not available for download in order to protect the identity of reviewers who chose to remain anonymous.

Reviewer 2 ·

Basic reporting

No comment

Experimental design

No comment

Validity of the findings

No comment

Additional comments

No comment

·

Basic reporting

No comment.

Experimental design

No comment.

Validity of the findings

No comment.

Additional comments

I have just the recommendations (personal point of view):
1) It is not necessary to use so thick scale bars and the letters used in figures seem to large in the ratio of the photo/drawing area.
2) Dataset is reduced, but I mean, for the confirmation of the new species is just enough to use sequences of the congeners and one or two close genera as the outgroup.

---

## Round 0.3 · accepted · Accept

After revising your manuscript, I am pleased to confirm that your paper has been accepted for publication in PeerJ.

Thank you for submitting your work to this journal.